# Spacing Dependent Mechanisms of Remagnetization in 1D System of Elongated Diamond Shaped Thin Magnetic Particles

Dominika Kuźma *, Oleksandr Pastukh and Piotr Zieliński *

Institute of Nuclear Physics Polish Academy of Sciences, PL-31342 Krakow, Poland
* Correspondence: dominika.kuzma@ifj.edu.pl (D.K.); piotr.zielinski@ifj.edu.pl (P.Z.)

**Abstract:** Four different switching scenarios have been revealed for a linear chain of flat magnetic particles with long axes perpendicular to the axis of the chain. The diamond-like shape of the particles has been previously shown to be the best to ensure a uniform and stable magnetization within a single particle, that is, to behave as a macrospin. The occurrence of each scenario depends on the distance of the particles in the chain. Whereas long distances favor direct remagnetization of a ferromagnetic configuration "all up" to the one "all down", a short enough distance allows the system to recover its ground state at a zero field, that is, an antiferromagnetic order. This allows any information stored by a magnetic field to be erased. Vortex-like metastable defects have been noticed for intermediate distances. A longitudinal magnetization component at extremely short distances has been noticed as well as specific systems of domain walls. The hysteresis loops and magnetization maps in the particles have been presented for each scenario. The potential applicability of the findings to the fabrication of memory storage devices has been discussed.

**Keywords:** macrospin; micromagnetic simulations; remagnetization; memory storage devices; tensile stress

## 1. Introduction

Nanometer-sized magnetic particles gain nowadays the growing attention of researchers and engineers due to their unique electric and magnetic properties enabling them to be used for future technological applications. Advances in fabrication methods (such as high-resolution lithography [1–4], electrochemical methods [5–7] and others [8–10]), make it possible to control the size, shape and spatial arrangement of magnetic nanostructures and, therefore, to obtain materials with tunable magnetic properties desirable for technological purposes [11].

Of special interest are monodomain ferromagnetic particles with elongated morphology, such as cylinders, prolate ellipsoids and stripes. The shape anisotropy favors single-domain stable magnetization in such particles which consequently can be considered and used as macrospin objects [12]. Since the magnetization of such bistable nanomagnets can be switched by an external magnetic field, the nanomagnets are promising candidates for high-density magnetic memory devices [13–15], as well as magnonic constituents of telecommunication systems. Therefore, precise quantitative knowledge is needed of the equilibrium states and the reactions of the appropriately designed macrospin systems to variations of the applied magnetic field. The latter is in fact the means of recording and reading information. The dominating interactions in systems of macrospins are dipolar. They may affect the magnetic moment distribution inside each macrospin. Consequently, the switching process may significantly differ from a simple coherent reorientation [16]. In addition, the remagnetization of neighboring macrospins can be a complicated process involving non-synchronous switching of adjacent regions, which influences the coercivity and squareness of the hysteresis loop [17]. Therefore, the optimization of shape, dimensions and spacing between magnetic particles is crucial for their correct performance as elements of memory storage and telecommunication devices [18–20].

In previous studies, we have shown that the switching mechanism of elongated magnetic nanoparticles made of permalloy strongly depends on whether the nanoparticles are narrowed or expanded at their ends [21]. It has turned out that a well rectangular hysteresis loop with almost monodomain magnetization in the whole volume was achieved for the elongated diamond-like shape structures. The switching process depends on the mutual situating of macrospins because it determines the impact of dipolar interactions on the overall magnetization as well as on its distribution within each macrospin. In a linear chain of such macrospins, the decisive role is played by their distance. It affects the coercivity and the squareness of the hysteresis [22] but it can also affect the magnetic switching mechanism [23]. It was shown, that the remagnetization process of an array of long magnetic stripes of a small (submicron) thickness can change from uncorrelated to cooperative switching as a function of the stripes' distance [20]. The use of micromagnetic simulations offers a possibility of modelling such systems and brings new knowledge concerning the dipolar coupling between magnetic objects [24,25].

In the present study, we consider a linear chain of magnetic nanoparticles of flat, quasi-2D shapes. Such nanoparticles are relatively simple to fabricate with nanolithographic techniques [4] and previous literature discusses their potential applications, for example in memory storage devices [21,26]. Due to their elongated shape, such magnetic nanoparticles tend to behave like single spins and therefore, can be considered as macrospins. However, we show that at some interparticle distances internal inhomogeneities come into effect. The resulting system of domains and domain walls may be advantageous for the fabrication of memory and/or telecommunication devices [27].

The shapes of the particles considered here are elongated in the direction perpendicular to the chain. The shape anisotropy then favors magnetization perpendicular to the chain which is particularly advantageous in designing memory devices. The shape of each particle stems from a diamond with an elongated $y$-axis. This shape has been proven to show the best tendency to a uniform magnetization in each particle, that is an enhanced macrospin behavior even in the absence of magnetocrystalline anisotropy. At the same time, the system exhibits the largest and well-squared hysteresis out of the shapes studied [21]. On the other hand, the central parts introduce proximity of the magnetic spins of the neighboring particles that give rise to regions where the shape anisotropy may favor a longitudinal magnetization, significantly modifying the switching properties. Thus, the interparticle distances are crucial for the magnetic behavior of the system. The distances can be well fixed if the non-magnetic substrate is sufficiently rigid, but the most interesting case is that of elastic deformable substrates. The interparticle separation may then be tuned with applied stress. Stretchable nanocomposites containing single-domain magnetic nanoparticles have been shown to manifest their strain (mechanical deformation) by a change in the specific absorption rate of energy when subjected to a magnetic field in the radio range of frequency (3 kHz–300 GHz) [28]. The particles' magnetization has been, however, reduced to point-like magnetic moments (macrospins) interacting via, Zeeman, dipole-dipole and anisotropy forces. In our work, we consider a collection of magnetic particles on a verge between the nanometric and micrometric size in order to study the detailed mechanisms of remagnetization involving controlled intraparticle distance. The shape of the particles has been selected so as to ensure the most pronounced macrospin behaviour due to the shape anisotropy without any contribution of a magnetocrystalline anisotropy. We reveal a cross-over between a direct remagnetization, that is, a transition between opposite ferromagnetic orders and a three-step remagnetization with an intermediate antiferromagnetic configuration being the systems' ground state at the zero external field. Our system is also a detector of mechanical deformation this time in a quasistatic external field, the signature of deformation being the coercive field with a significant cross-over to a three-state mechanism for sufficiently squeezed geometry. Stress tuneable systems are of interest in modern technology for example durable materials such as tires, smart actuators and sensor materials [29].

## 2. Methods, Materials and Geometry of the System

The magnetic state of a nanoparticle or a collection of such particles is defined with a magnetization map indicating the strength of the magnetic moment density considering the whole volume. Here we use the MuMax3 v.3.10 (DyNaMat group, Ghent University, Ghent, Belgium) software [30,31] to obtain interesting magnetization maps with micromagnetic computations in which every particle is divided into cubic voxels of size 5 nm × 5 nm × 5 nm. We consider permalloy ($Ni_{80}Fe_{20}$) as an example of a classical material without magnetocrystalline anisotropy. The saturation magnetization and exchange stiffness constant of permalloy are equal to $M_s = 860 \frac{kA}{m}$, $A = 1.3 \times 10^{-11} \frac{J}{m}$, respectively [32,33]. The only contribution to the anisotropy is shape anisotropy [34–36]. The sizes of the elongated diamond-like nanoparticles studied here are given by the parameters visible in Figure 1: $S_x = 350$ nm, $S_y = 5000$ nm, $S_z = 30$ nm, where the subscripts $x, y, z$ correspond to perpendicular directions: along the chain, along the long macrospin axes and perpendicular to the particles' plane respectively. The thickness of the macrospins assumed to be 30 nm, allows the magnetization to lie practically within the $x$, $y$ plane due to the geometry of the demagnetization field. The length of our particles is rather large, to ensure good macrospin behavior. On the other hand, the size ensures a relative smoothness of the particles, that is, a roughness due to the computation voxels is negligible. Despite the micrometer length, particles of this size are usually called nanoparticles [37].

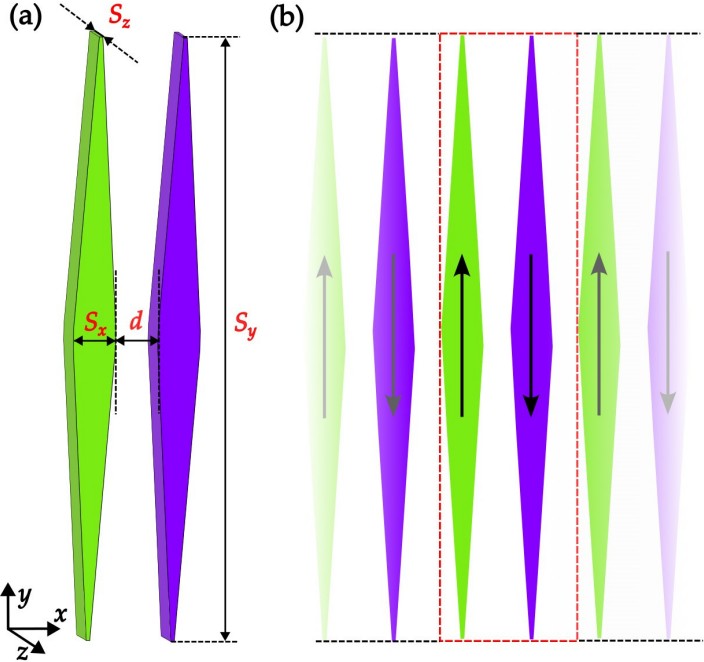

**Figure 1.** (**a**) Schematic representation of a pair of parallel elongated diamond-shaped nanoparticles. The distance *d* concerns the lateral apices of the diamonds. (**b**) Sequence of considered nanoparticles arranged in a linear chain. The arrows indicate magnetization directions within each object in the ground state.

The magnetization maps have been visualized with tools provided by the MuMax3 software. Figures showing magnetization maps have been obtained in this way. The software MuMax3 also allows the user to extract the resultant magnetization of every particle. This is particularly useful when considering the particle as a macrospin, that is, a carrier of a net magnetic moment. The evolution of such resultant magnetic moments in 1D arrangements considered here is presented described in the next section. The contributions of different parts of energy have been studied to determine the scenarios of the switching processes. An example of the total energy dependence on the applied field in stable and metastable equilibrium states is presented in Figure 2. The obtained configurations are

the results of searches for minimum, global or local, performed by the software MuMax3. The results are slightly dependent on the adopted step in the applied magnetic field. We show in Figure 2 the extent of the differences exemplified with three different steps. We have routinely used the field step equal to 0.01 mT. The main problem solved here is the dependence of the magnetic equilibrium configurations on the distance $d$ between the neighboring macrospins in the chain as illustrated in Figure 1. Due to the configurations with parallel moments of the particles (FM, or "all up" when upwards and rFM or "all down" when downwards) and with antiparallel (AF) macrospins order are the most interesting for memory device design, we impose periodic boundary conditions limiting the repetition period to two neighboring macrospins.

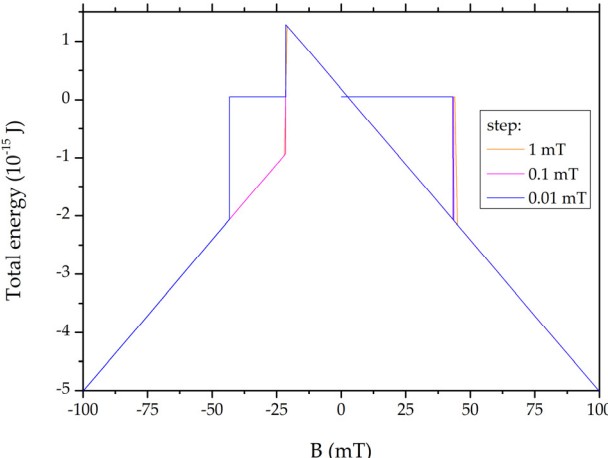

**Figure 2.** Total energy of different stable and metastable configurations of the chain at the distance $d = 10$ nm. The minima found by the software MuMax3 are slightly dependent on the field step $\Delta B$, which has also been shown for three selected values.

## 3. Results

The most stable configuration in the absence of an external field is the antiferromagnetic (AF) one. This is a consequence of the dipolar interactions favoring head-tail neighbors in the present geometry. A competitive tendency of parallelization of the microspins (voxels), especially those close to the $x$-axis, comes into effect with decreasing distance $d$. Nevertheless, a quasi-antiferromagnetic configuration with a non-zero $x$ component of magnetization has been proved to have the lowest possible energy even in the limiting case of $d = 0$ (Figure 3). Consequently, we assume the AF configuration as the starting state for the thought experiment with a variable external magnetic field applied in the $y$ direction.

In the limit case of $d = 0$ we have checked the final magnetization maps starting from some selected initial configurations. The results are presented in Figure 3. The perfect antiferromagnetic configuration relaxes to the most stable of all the configurations found (Figure 3a). It remains similar to the initial one except for the central part where a zig-zag (upwards right–downwards right) deviation is visible. This is different from the known domain wall structures. However, when looking along the path starting from the bottom left and going through the middle to the top right, the structure may be reminiscent of a 180° transverse domain wall (TDW, see [22,38]). One can see two such interpenetrating TDW in Figure 3a. A typical 180° TDW (see [38,39]) in the right diamond of Figure 3c seems to be stabilized by a deviation of the central voxels in the left diamond. A similar domain structure has been found in an H-shaped system (Figure 2b of the reference [37]). A more involved domain structure has been obtained starting from a random spin distribution in Figure 3d.

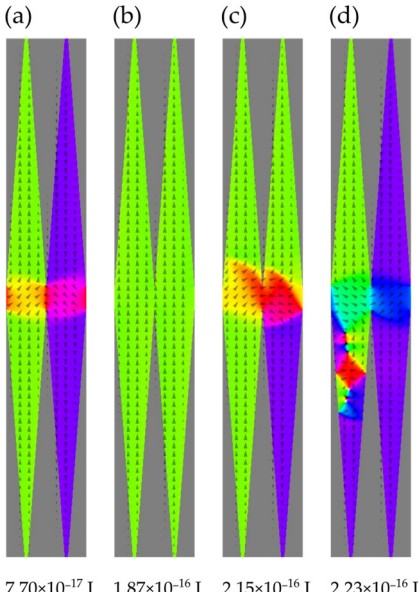

(a)  (b)  (c)  (d)

7.70×10⁻¹⁷ J    1.87×10⁻¹⁶ J    2.15×10⁻¹⁶ J    2.23×10⁻¹⁶ J

**Figure 3.** Relaxed magnetization maps for sticking diamonds, i.e., for $d = 0$ obtained at $B = 0$ by starting from (**a**) perfect AF, (**b**) perfect FM, (**c**) with all spins parallel to $x$ and (**d**) randomly oriented spins. The total energy of the relaxed configuration is given below each panel.

The behavior of the total magnetization in the system under an external magnetic field parallel to the long particles' axes for a number of distances $d$ is depicted in Figure 4. For comparison, the magnetization of a single diamond-shaped particle is represented with a continuous solid orange line. It has been found [18,21] that the hysteresis is the largest possible in this case. The presence of neighboring macrospins, thus, facilitates the reversal process. Moreover, the particle then remains uniformly magnetized at all values of the field. This manifests itself by a well-squared shape of the hysteresis loop. The other curves of Figure 4 correspond to decreasing distance $d$ starting from 350 nm.

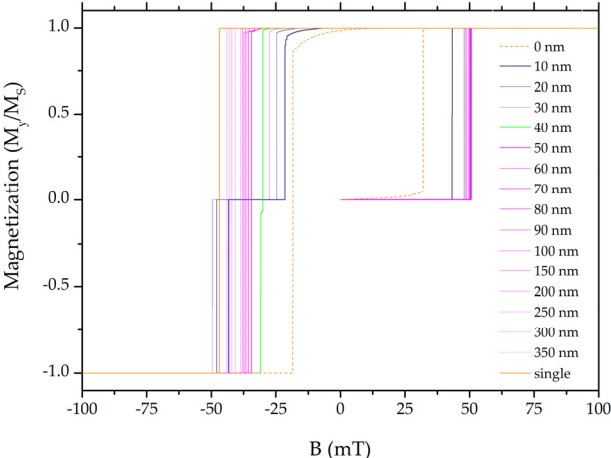

**Figure 4.** Evolution of the magnetization parallel to the long axes of the elongated diamond-shaped macrospins with varying intensity $B$ with the external magnetic field for selected interparticle distances $d$. The starting point at $B = 0$ is the AF configuration. The largest hysteresis occurs for a single nanoparticle (FM→rFM). Similar colors correspond to particular types of switching behavior: magenta FM→rFM, green FM→AF→rFM and navy FM→dAF→rFM. The dashed orange line corresponds to the distance $d = 0$.

Every simulation cycle starts with the equilibration (relaxation) of the AF configuration of initially uniformly magnetized particles at $B = 0$. The resulting equilibrated maps turn

out fairly uniform too except for the limit case $d = 0$. This uniform magnetization persists up to the AF→FM switching as can be seen from the left panel (a) in Figure 5 which is an example of this behavior for $d = 60$ nm. An abrupt reversal is observed to the configuration FM, that is, "all up" also with no discernable inhomogeneities (see panel (b) in Figure 5). Generally, the switching field of the AF→FM reversal increases with increasing interparticle distance $d$.

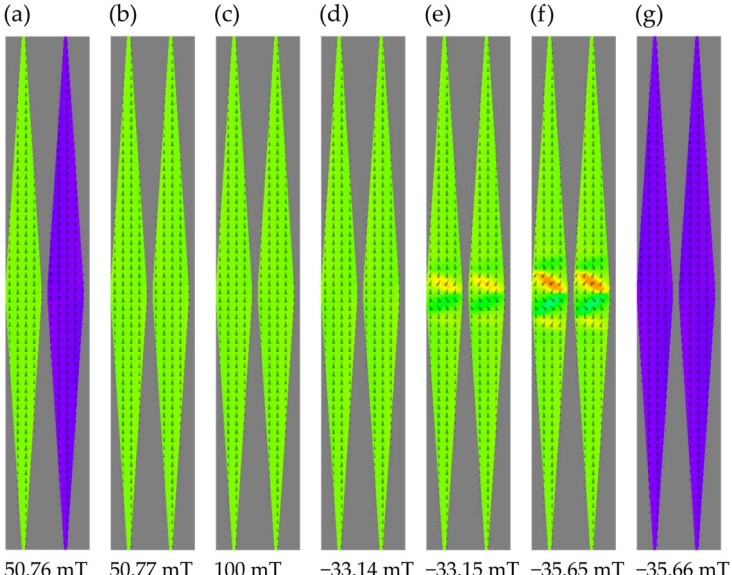

**Figure 5.** Magnetization maps for a pair of macrospins at the separation distance $d = 60$ nm. (**a**,**b**) just before and just after the AF→FM switching, (**c**) at the largest applied field, (**d**–**g**) consecutive stages of the FM→rFM switching. Intensity of the applied field is given below each panel.

The AF→FM switching is more complex for the distance $d = 0$. As seen in Figure 6, the equilibrium magnetization map shows an important x-component in the region close to the chain's axis. This is a manifestation of the head-tail chain tendency of aligned dipoles. In other parts of the particles, however, the tendency to an antiparallel order gains as it should be for the dipoles perpendicular to a common axis. The configuration of the panel (a) of Figure 6 is the most stable one, as it shows the lowest total energy. For comparison, Figure 3 presents the magnetization maps corresponding to local energy minima obtained with some selected initial configurations.

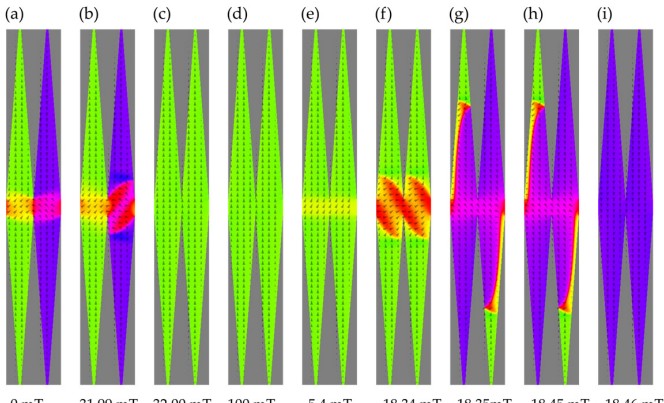

**Figure 6.** Magnetization maps for a pair of macrospins at the separation distance $d = 0$. (**a**) starting relaxed quasiAF configuration, (**b**,**c**) just before and just after the AF→FM switching, (**d**) at the largest applied field, (**e**) a precursor deviation in FM configuration (**f**–**i**) consecutive stages of the FM→rFM switching. Intensity of the applied field is given below each panel.

After attaining the "all up" FM configuration, the external field was increased to 0.1 T to produce a state corresponding to the physical effect of a strong field. This was the starting point for studying the reversal mechanism. Then the field was decreased to negative values to attain the "all down" configuration. The largest coercivity was visible for the most distant particles. This was consistent with the observation of the largest hysteresis for the single macrospin: an extremely large distance reduces the problem to the reversal of an isolated particle. The magnetic field was changed by a step of 0.01 mT for all systems studied.

The four following types of behavior have been observed:

1.  A direct FM→rFM, i.e., "all up" into "all down" switching occurs for a large enough (starting from 50 nm and higher) distance $d$. This is represented in Figure 4 with a magenta color. The reason for that is that the dipolar interactions of the neighboring diamonds are too weak to form the AF configuration. The value of the switching field decreases with decreasing distance $d$. Additionally, with the decreasing distance, the shape of the hysteresis loop becomes rounded which describes an inhomogeneity of the magnetization when approaching the switching value of the field. An example of inhomogeneity is represented in Figure 5 for $d = 60$ nm. The "all up" to "all down" remagnetization (FM→rFM) is preceded by a slight deviation in the central parts of the diamonds, as seen in Figure 5e,f. This is a zig-zag sequence reminiscent of but much lighter than the 360° domain walls. In contrast to the latter, there is no region of reversed spins. A similar structure has been called "S-state" in ref. [40].

2.  The green line in Figure 4 for $d = 40$ nm corresponds to an interesting case in which the initial FM configuration exhibits a deviation of the central parts of the macrospins so that an $x$-component of magnetization appears before switching. This contributes to a rounding of the hysteresis loop. Next, an antiferromagnetic configuration arises with a visible pair of vortices approximately halfway between the $x$-axis and the apex of the particle magnetized antiparallel to the field. The structure should not be confused with the "double vortex" one stabilized by ends of elongated particles with a circular cross-section (e.g., [41,42]). A single vortex of panels (c) and (d) of Figure 7 is reminiscent of an annihilating 360° domain wall [38]. This configuration with the pair of vortices has been given the name defected antiferromagnetic and denoted with dAF. The stage of switching corresponds to a small step on the green curve. Interestingly enough, the defect consisting of two opposite vortices shows a perfect compensation of chirality in that 180° rotation transforms it into itself (see Figure 7c,d). With further increases in the switching field, the defected antiferromagnetic configuration reverses into the rFM one with the magnetization parallel to the applied field. The consecutive stages of the switching process FM→dAF→rFM are shown in Figure 7.

3.  For even shorter distances, the system shows a two-step switching FM→AF→rFM, shown in Figure 4 with the navy curves. The range of distances is 10–35 nm. The switching process is initiated by precursor deviations resulting in a rounding of the hysteresis loop. The precursors are visible in Figure 8b.

4.  The limiting distance $d = 0$ corresponds to the sticking voxels belonging to neighboring particles. The configuration at $B = 0$ depends on the initial one in the minimization process. Figure 4 depicts the evolution starting from the AF one with the dashed orange line. As shown in Figure 6, the precursor deviations from the "all up" FM configuration are rather extended and exhibit a non-zero $x$ component of magnetization. By increasing the reversal field, a gradual reversal is observed, that is, an increase in the reversed region at the expense of the initial one. Panels (e) and (f) of Figure 6 show developing coupled 180° domain walls that dissociate into decoupled 180° domain walls in each diamond [43,44].

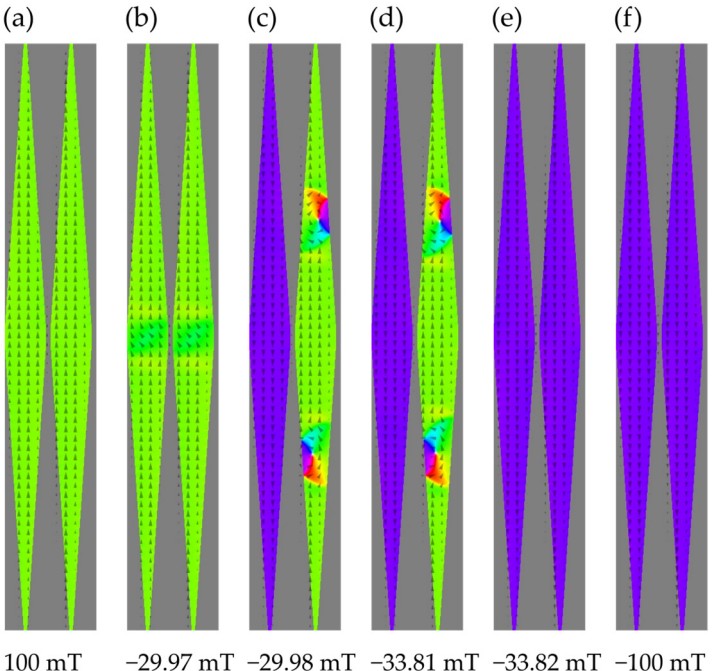

100 mT    −29.97 mT  −29.98 mT  −33.81 mT  −33.82 mT  −100 mT

**Figure 7.** Switching process FM→dAF→rFM at the distance $d = 40$ nm. (**a**) saturated FM (**b**) visible precursor deviation of magnetization in the central part, (**c**,**d**) left diamond entirely reversed, pair of vortices in the right diamond, (**e**) configuration rFM just after reversal, (**f**) saturated rFM at the largest negative applied field.

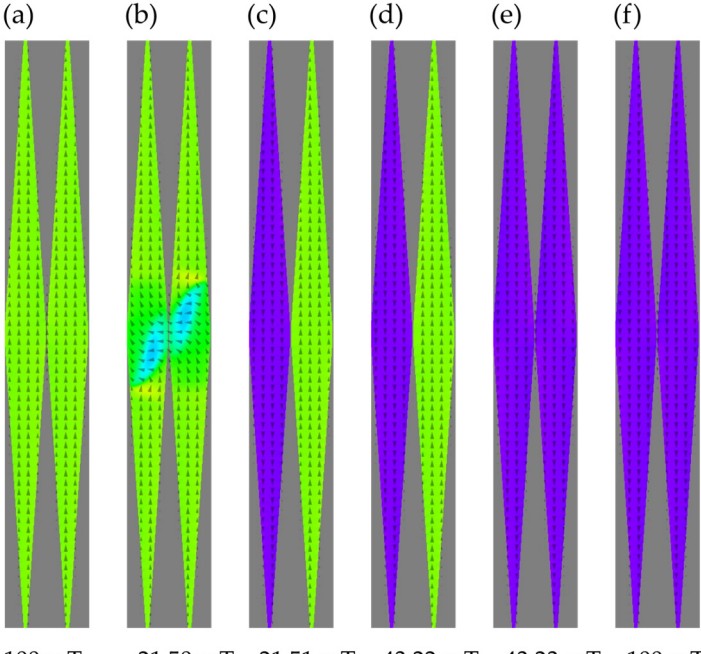

100 mT    −21.50 mT  −21.51 mT  −43.22 mT  −43.23 mT  −100 mT

**Figure 8.** Magnetization maps for a pair of macrospins at the separation distance $d = 10$ nm. (**a**) saturated FM configuration, (**b**,**c**) just before and just after the FM → AF switching, (**d**,**e**) just before and just after the AF→rFM, (**f**) saturated rFM configuration.

## 4. Discussion and Conclusions

The present computations have demonstrated that a chain of the particles shaped so as to exhibit optimal macrospin properties shows two general types of switching behavior as a function of the interparticle distance. The long-distance behavior involves a remag-

netization (FM→rFM) between two oppositely magnetized states. Thus, the reaction to the variations of the external applied field reproduces the one for a single macrospin, with only a narrower hysteresis loop and a reduced coercivity. If the distance $d$ is short enough, here $d \leq 35$ nm, a new quality comes into effect, that is, a return to the AF configuration which is the most stable one at $B = 0$. Figure 2 depicts the total energy as a function of the applied magnetic field. This is particularly promising since the configuration AF corresponds to a state "all erased" or "virgin". The memory element may then be reused for further operations. Two additional kinds of behavior have been observed. A defected dAF configuration has been found for a narrow region close to $d = 40$ nm. The configuration is marked with two oppositely oriented vortices. Such objects, when well stable, may also carry information on internal details of magnetization in a single particle, then different from a macrospin, could be written/read by the appropriate devices.

　　　　Another limiting case is the one of sticking diamond-shaped particles: $d = 0$. The system then shows the narrowest hysteresis with the FM→rFM switching. A particularity of this case resides in a fairly rounded hysteresis and a significant well-polarized contribution from the $x$-component of magnetization absent from the defected configurations mentioned above. Of course, one can think of devices capable of detecting this $x$-component. A practical question may arise of how to fabricate a system exhibiting the properties described. An ideal case would involve the adsorption of the nanoparticles on a very elastic distensible substrate that could change its length under external tensile stress without affecting the shape of the particles. Whereas Dudek et al. [28] suggested using the radiation energy absorption as a detector of mechanical deformation, here we have shown that the extent of hysteresis along with the cross-over between the one-step and two-step switching may be indicative of tensile strain. Of practical interest are also intraparticle inhomogeneities of magnetization reported in Figures 3c, 5f, 6a,b and 7c,d.

**Author Contributions:** Conceptualization, D.K. and P.Z.; methodology, D.K. and P.Z.; software, D.K.; visualization, D.K., O.P.; supervision, P.Z.; investigation, D.K., P.Z., O.P.; writing—original draft preparation, D.K., P.Z., O.P.; writing—review and editing, D.K., P.Z., O.P. All authors have read and agreed to the published version of the manuscript.

**Funding:** The numerical calculations were performed at Poznan Supercomputing and Networking Center (Grant No. 424).

**Institutional Review Board Statement:** Not applicable.

**Informed Consent Statement:** Not applicable.

**Data Availability Statement:** All data generated or analysed during this study are included in the manusript.

**Conflicts of Interest:** The authors declare no conflict of interest.

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
