# Peer review of "Spacing Dependent Mechanisms of Remagnetization in 1D System of Elongated Diamond Shaped Thin Magnetic Particles"

_magnetochemistry, doi:10.3390/magnetochemistry8090102_

Round 1

Reviewer 1 Report

In this work Kuźma et al., present the theoretical evaluation of four different switching scenarios  for a linear chain of flat magnetic  particles elongated perpendicularly to the chain axis, which behaves as a macrospin. The dependance of the distance between the particle chain is the fundamental property to differentiate each scenario. The hysteresis loops and the magnetization maps in the particles have been presented for each scenario. Although I support the publication of this work in Magnetochemistry, I have just three main details that the authors need to adress

The english need to be improved, there are some ideas along the text that are not understandable.

There is anyway that the authors can provide real experimental data and not only a theoretical approach

Can the authors provide more evidence about the potential applicability of these findings for the fabrication of memory storage devices.

Reviewer 2 Report

The authors report the magnetic switching of thin magnetic nanoparticles with variable distance. Anyway, it is nice to understand the magnetizaion reversal with the external magnetic field.  Here I hope the authors can give more detials about their work's application. If their work can apply, which field can be applied? Also, the application standard and advantages when their magnetic structure can be applied. 

Reviewer 3 Report

The manuscript deals with micromagnetic study of chains of elongated magnetic particles/wires of Permalloy. The authors discuss switching mechanisms performing relaxation of the magnetization with dependence on the distance between nanoparticles. Results include hysteresis curves, which change their character between ferromagnetic to antiferromagnetic ones with changing the strenght of the magnetostatic coupling. The results look trustworthy, while,  the structures related to particular reversal modes of magnetic nanowires should be classified by means of naming characteristic textures. In particular, in the snapshot of Fig. 3(c), one can recognize the well-known transvere domain wall. In Figs 5(f), 6(a), (b), (f) there are probably two coupled 360' transverse domain valls, (vortex or transverse ones), e.g. A. Kunz APL 94, 132502 (2009), D. Djuhana et al. JAP 106, 103926 (2009). Figs. 7(c), (d) present a pair of 360' domain walls. They shouldn't be called double vortex, which is reserved to short nanowires and stabilized be the ends of the nanowire/nanoparticle (e.g. S. S. Cherepov et al. PRL 109, 097204 (2012), H. Hata PRB 90, 104418 (2014), A. Janutka, IEEE Magn. Lett. 10, 6103105 (2019).

In my opinion, there is a major weakness of the manuscript in terms of the motivation description as well as in the title. In terms of the title, I think that naming 5000nm long and up to 500nm wide magnetic systems nanoparticles is misleading. They are closer to nanowires than nanparticles. On the other hand, the authors motivate their work with the application of chains of the elongated particles/wires to the magnetic memory/storage. I don't think relatively big particles under consideration to be of interest as memory bits at present standards of the bit miniaturization.

Round 2

Reviewer 3 Report

The authors have taken into account my comments and improved the manuscript. I still see some points to be improved, however.

In page 5, bottom, there is a reference to Figure 2(b), while, Figure 2 is not divided into parts.

In pages 8-9, the description of Figures 7(c), 7(d), I don't think the structures seen can be called votices. They are 360deg domain walls rather.
